# Internet Healthcare Policy Analysis, Evaluation, and Improvement Path: Multidimensional Perspectives

**DOI:** 10.3390/healthcare11131905

**Published:** 2023-06-30

**Authors:** Qi Wei, Xiaoyu Wang, Gongrang Zhang, Xingguo Li, Xuejie Yang, Dongxiao Gu

**Affiliations:** 1School of Management, Hefei University of Technology, Hefei 230009, China; 2021110918@mail.hfut.edu.cn (Q.W.); grzhang118@126.com (G.Z.); lixingguo@hfut.edu.cn (X.L.); xuejie_y@mail.hfut.edu.cn (X.Y.); 2The Department of Pharmacy, Anhui University of Traditional Chinese Medicine, Hefei 230009, China

**Keywords:** internet healthcare, policy text analysis, policy instruments, policy evaluation

## Abstract

Internet healthcare is a crucial component of the healthcare industry’s digital transformation and plays a vital role in achieving China’s Healthy China strategy and promoting universal health. To ensure the development of internet healthcare is guided by scientifically sound policies, this study analyzes and assesses current policy texts, aiming to identify potential issues and inadequacies. By examining 134 national-level policy documents, utilizing multiple research methods, including policy bibliometrics, content analysis, and the PMC Index Model, the study investigates policy characteristics, distribution of policy instruments, and evaluation outcomes related to internet healthcare. The study findings reveal that internet healthcare policies place emphasis on enhancing service quality, driving technological innovation, and promoting management standardization. Although policy instruments align with the current stage of internet healthcare development in China, they are plagued by imbalances in implementation. While policies are generally well-formulated, there are discernible discrepancies among them, necessitating the reinforcement and refinement of certain provisions. Hence, it is imperative to strategically optimize the amalgamation and implementation of policy instruments while concurrently endeavoring to achieve a dynamic equilibrium in policy combinations. Furthermore, policymakers should diligently refine the policy content pertaining to its nature and effectiveness in order to fully maximize policy utility.

## 1. Introduction

Internet healthcare refers to a new medical service model that deeply integrates the internet as a carrier and technological means with the healthcare industry, allowing for a more efficient and convenient provision of medical services and health management [1]. With the rapid development of online platforms and digital technologies, Internet healthcare has emerged as a promising avenue to enhance healthcare accessibility, efficiency, and public health outcomes [2,3].

In response to this trend, since 2014, the Chinese government has introduced a range of policy documents, including implementation opinions, development plans, regulatory constraints, and medical insurance payment policies, to support and guide the growth of the internet-based healthcare industry. Based on the characteristics of China’s internet healthcare development and practice, as well as the trend of policies introduced in different years, the development process of internet healthcare can be roughly divided into three stages.
(1)Initial construction phase (2014–2017): As healthcare demand increased and information technology continued to advance, the government implemented a set of policies during this period, such as the “Opinions on Promoting Telemedicine Services” (2014) and the “Guiding Opinions on Actively Promoting the ‘Internet Plus’ Action” (2015), aimed to stimulate the adoption of “Internet Plus” technologies in order to catalyze a significant transformation and growth within the healthcare sector.(2)Standardization establishment phase (2018–2019): In 2018, the State Council of China issued the “Opinions on Promoting the Development of ‘Internet Plus Healthcare’,” underscoring the imperative of standardization within the internet-based healthcare sector. The government placed significant emphasis on formulating comprehensive service standards and implementing a robust security system to facilitate the industry’s advancement throughout this phase.(3)Explosive growth phase (2020-present): The outbreak of the COVID-19 pandemic in early 2020 spurred the demand for internet-based healthcare. In response, the government issued a series of policies, including the “Notice on Strengthening Information Technology Support for the Prevention and Control of Novel Coronavirus Pneumonia Epidemic” (2020). These policies covered various areas, such as the construction of smart hospitals, internet-based healthcare insurance services, and online sales of prescription drugs, and aimed to support and regulate the development of the internet healthcare industry.

In the dynamic and transformative landscape of internet healthcare, it is essential for internet healthcare policies to be designed in alignment with the current state of industry development. This enables real-time adjustments to guide and regulate the industry effectively. Reviewing the design and implementation of current internet healthcare policies in China, what are the overall characteristics of these policies? Have the policy instruments and combinations been appropriate? Is the policy design comprehensive and sound? Policy text serves as an objective record of policy intentions and processes [4]. Analyzing policy text is crucial for tracing and observing policy processes, addressing policy concerns, and ensuring the scientific and rational nature of policies [5].

Therefore, the primary objective of this research is to undertake a comprehensive analysis and evaluation of China’s internet healthcare policies by employing methods such as policy bibliometrics, content analysis, and the PMC Index Model to conduct a systematic and multidimensional textual analysis and evaluation of 134 internet healthcare policies issued at the national level between 2014 and 2022. The analysis aims to extract the policy characteristics, distribution of policy instruments, and evaluation results pertaining to internet healthcare and analyze their characteristics and existing issues across three dimensions. The findings suggest that China’s internet healthcare policies suffer from imbalanced use of policy instruments, weak complementarity, and inadequate policy design. Consequently, targeted recommendations will be formulated to enhance the effectiveness of China’s internet healthcare policies, with the ultimate goal of promoting a robust and well-regulated development of internet healthcare.

The remainder of this paper is organized as follows. Section 2 reviews the related literature. Section 3 introduces policy samples and methodology. Section 4 presents the empirical results. Section 5 interprets the finding and concludes this paper.

## 2. Literature Review

As internet healthcare gradually enters the public’s vision, research on internet healthcare policies has become a hot topic of academic concern. Based on the existing research literature, studies on internet healthcare policies can be divided into three aspects.

The first category of research focuses on policy development systems in the field of internet healthcare. Researchers primarily examine the current state of development in internet healthcare and analyze the evolution of internet healthcare policies [6]. Silva et al. [7] conducted an analysis of the relevant regulations in internet healthcare and provided a description of the historical evolution of Brazilian internet healthcare policies across three periods. Ortega et al. [8] analyzed the specific clauses within internet healthcare policies, identifying the problems and challenges encountered during policy implementation and proposed strategies for adjusting internet healthcare services to better meet patient needs. Ekeland et al. [9] utilized qualitative research methods to discuss approaches for achieving quality improvement and necessary digital transformation in the governance of internet healthcare. Huang et al. [10] conducted an evaluation of health informationization in China, considering factors such as health information infrastructure, information technology application, financial and intellectual investment, health resource allocation, and standard systems. Gu et al. [11] employed bibliometric methods to examine the evolution of electronic health and remote medical care, providing insights into the knowledge structure and research hotspots from a global perspective. These studies contribute to a better understanding of the current trends and challenges that researchers face in the field of internet healthcare.

The second area of focus is evaluating the effectiveness of internet healthcare policies. Internet hospitals, which utilize internet technology to provide medical services, are a crucial component of the internet healthcare field. Therefore, most scholars assess the impact of internet healthcare policies by studying internet hospitals. For example, Xie et al. [12] provided an overview of Chinese internet hospitals and evaluated their medical service capabilities. Han et al. [13] summarized the definition and current status of internet hospitals and analyzed the construction and content of Chinese internet hospitals. Zhi et al. [14] used the case of the Zhejiang University Internet Hospital to analyze the operational mode and functional positioning of internet healthcare services. Ouchani and Gholami et al. [15,16] constructed concrete models of hospital management systems using model-based design methodologies to facilitate the integration and communication of medical resources. Lai et al. [17] examined the impact of policy intervention on the development trends and service innovation of internet hospitals through literature analysis and qualitative interviews. Xu et al. [18] analyzed the situation of Chinese internet hospitals during the COVID-19 pandemic, as well as the medical prescription, drug transportation, and medical insurance services they provided.

The third aspect involves the analysis of factors that influence internet healthcare policies and the exploration of relevant facilitating factors for their implementation. Bansal et al. [19] conducted a study that revealed the significant impact of users’ trust in internet healthcare platforms and concerns about privacy on their willingness to utilize such platforms. These factors, in turn, influence the effectiveness of implementing internet healthcare policies. Avanesova et al. [20] identified the critical roles played by national policies, medical capabilities, and infrastructure construction in facilitating the successful implementation of internet healthcare. Luciano et al. [21] identified policies, cultural aspects, and factors related to safety and privacy as important influencers on the implementation of internet healthcare. Powell et al. [22] examined users’ motivations and the facilitating factors that contribute to the utilization of medical services through internet healthcare platforms, and they found that different health behaviors can arise based on various directions of internet healthcare use and users’ access to online health information [23]. Wang et al. [24] proposed that policymakers should prioritize efforts to improve internet access and usage for individuals residing in rural areas, thus bridging the digital divide between urban and rural residents.

Overall, scholars have a theoretical basis for their research on internet healthcare policies. However, the current research lacks depth and systematic analysis (as shown in Table 1). The existing frameworks for policy analysis are primarily based on a single perspective and level, lacking comprehensive and multidimensional research on the internet healthcare policy system. Therefore, this study aims to construct an analysis framework for internet healthcare policies, encompassing “policy topics—policy instruments—policy evaluation.” This study combines qualitative analysis of policy text content with quantitative evaluation of policy implementation effects, taking a multidimensional perspective. The goal is to provide practical and feasible reference guidelines and standards for the design and improvement of future internet healthcare policies.

## 3. Materials and Methods

### 3.1. Research Materials

Considering the hierarchical nature of policy formulation, with local policies being guided by central policies, national policies have a broader influence and representativeness. Furthermore, national policies often embody universal experiences and general principles applicable across different regions. In our analysis, we only selected national-level policy documents on internet healthcare, recognizing their significance in capturing the overarching trends and guiding principles within the field. Multiple data sources were used, including the PKULAW database and websites of relevant departments, such as the State Council and the National Health Commission, with search terms such as “internet healthcare”, “internet hospital”, and “internet health”. Due to the fact that the practical exploration of internet healthcare preceded policy formulation before 2014, the presence of policies during that period was relatively limited. Since 2014, the government has exhibited an increasing level of support and promotion for internet healthcare, resulting in the implementation of a series of significant policies and regulations in the subsequent years, which have generated a sustained impact on the development of internet healthcare. Therefore, the policy retrieval period was limited from 1 January 2014 to 31 December 2022. In order to ensure that policy content is relevant to the theme of internet healthcare, the policy texts were filtered and screened according to inclusion and exclusion criteria, as outlined in Figure 1.

After the screening, a total of 134 valid national policy documents were obtained and numbered in chronological order, as shown in Appendix A.

### 3.2. Research Design

In order to achieve a scientifically rigorous quantitative evaluation, this study situates the policy paradigm within the context of China’s internet healthcare policies. The analysis of policies is broken down into key elements such as policy issues, policy objectives, and policy instruments. A framework of “policy keywords—policy instruments—policy evaluation” is constructed, as outlined in Figure 2. This framework encompasses the comprehensive application of three perspectives.

Firstly, topic keywords were used to represent the content characteristics of policies. Analyzing policies based on topic keywords allows for a reflection of the policy’s core content objectives or issues of the policies, allowing for a better understanding of the overall features and focus of internet healthcare-related policies. It enables a more precise and focused examination of the comprehensive features and focal points of internet healthcare-related policies, thereby enhancing the specificity and targeted nature of the analysis. Thus, this study employed a policy topic keyword perspective and utilized the policy literature’s bibliometrics to conduct word frequency statistics and semantic network analysis on the topic keywords [25], which led to the discovery of the policies’ basic features and points of concern.

Secondly, policy instruments are the tangible manifestation of public policy design and selection, representing the specific means or methods utilized by governments to achieve policy objectives [26]. By analyzing the selection and utilization of policy instruments, it becomes possible to assess the feasibility and effectiveness of policies. This analysis aids in comprehending how policies can purposefully influence the development of internet healthcare. Therefore, this study analyzed the supply, demand, and environmental characteristics of China’s internet healthcare policies using content analysis from the perspective of policy instruments. It dissected the issues of internal balance and matching within policies and interpreted the government’s attention allocation features.

Moreover, policy evaluation represents a crucial component within the process of public policy formulation and management. It involves a comprehensive examination and analysis of policies using scientific evaluation criteria and methods. This process enables not only the scientific assessment of the inherent value of policies but also the examination of the actual effects of policy formulation and implementation [27]. Its significance lies in facilitating the development, execution, and iterative adjustments of internet policies [28]. Hence, this study constructed a PMC index evaluation model from the perspective of policy evaluation to identify deficiencies in policy design across various dimensions and achieve an overall evaluation of the effectiveness of policy implementation.

By integrating the aforementioned three perspectives, a more comprehensive analysis can be conducted to better clarify the core content and objectives of policies, assess the strengths and limitations of policies, and evaluate their feasibility and effectiveness. Lastly, drawing from the systematic multidimensional analysis and aligning with China’s national strategic goals for developing internet healthcare, this study presents targeted recommendations to enhance the effectiveness of internet healthcare policies.

### 3.3. Research Method

#### 3.3.1. Policy Bibliometrics

Policy bibliometrics, as a branch of bibliometrics, is a research method that systematically analyzes and evaluates the structural attributes of policy literature [29]. Its purpose is to uncover the dynamics and trends within the field of policy research by employing quantitative analysis of variables such as the quantity of the policy literature, keywords, authors and institutions, and citation relationships, among others [30]. This approach aims to provide decision support for policy formulation and implementation by shedding light on the evolving landscape of policy studies. Inspired by the principles of bibliometric research pertaining to keywords, this study relied on policy keywords and employed various techniques, including word frequency analysis, co-occurrence analysis, and network analysis, to identify highly specific policy keywords and their interrelationships. Furthermore, it aimed to explore the distinctive characteristics of policies and discern emerging trends within the field.

#### 3.3.2. Policy Instruments

Policy instruments are essential instruments employed by governments to accomplish their policy objectives through a series of measures. Scholars have approached the classification of these instruments from various perspectives. For instance, Howlett et al. [31] categorized policy instruments as voluntary, hybrid, or coercive, depending on the extent of government intervention. On the other hand, McDonnell et al. [32] classified them as incentive-based, command-based, learning-based, or systemic-based, considering the government’s objectives. Rothwell and Zegveld [33] proposed a classification scheme that distinguishes policy instruments as supply-side, environmental, or demand-side based on their impact on technological activities—a widely acknowledged and applied categorization. In this article, we adopted Rothwell and Zegveld’s classification method and organized policy instruments in the domain of internet healthcare into supply-side, demand-side, and environmental types. Moreover, each type is further subdivided into specific policy instruments tailored to the characteristics of internet healthcare, as outlined in Table 2.

#### 3.3.3. PMC Index Model

Presently, widely employed approaches or framework models for policy evaluation, both domestically and internationally, include the Analytic Hierarchy Process (AHP), BP Neural Network Evaluation Method, Fuzzy Comprehensive Evaluation Method, Propensity Score Matching-Difference in Differences (PSM-DID) Method, Grey Relational Model, Dynamic Computable General Equilibrium (CGE) Model, and PMC Index Model, among others. However, many of these methods exhibit certain limitations when applied to policy texts, mainly stemming from inherent subjectivity and relatively lower precision.

The PMC (Policy Modeling Consistency) Index Model, proposed by Ruiz Estrada [34], is a policy evaluation tool that combines traditional text mining methods with advanced mathematical instruments to construct a quantified evaluation model. This model focuses on the raw data of policy texts, aiming to incorporate relevant variables as comprehensively as possible, considering various factors. By assessing the internal consistency of individual policies and enabling comparison among multiple policies, the model effectively embodies the scientific, fair, and rational aspects of the public policy-making process. The analysis process consists of three steps: variable and parameter setting, PMC index calculation, and PMC surface construction.

The first step involves the determination of variables and parameter settings. In this study, reference was made to the research conducted by Ruiz Estrada [35] on policy evaluation, along with relevant explorations [36,37,38,39,40], to identify the criteria for policy evaluation. A total of ten primary variables were selected. Subsequently, taking into consideration the secondary variables discussed by the aforementioned scholars, as well as the analysis of internet healthcare policy text content in Section 4.1 and 4.2, a comprehensive set of 38 secondary variables was established. These variables were based on the extracted policy thematic keywords and policy instruments, allowing for a more nuanced evaluation framework. In the parameter setting, all second-level variables are assigned equal weight, represented in a binary format, as illustrated in Table 3.

The second step involves calculating the PMC (Policy Modeling Consistency) index. Firstly, the binary values of the secondary variables are assigned one by one according to Equation (1). Secondly, the values of each primary variable are calculated according to Equation (2). Finally, the PMC index of each policy is calculated by substituting the values of all primary variables into Equation (3). The policy is then assigned an evaluation level based on Table 4.
(1)$$X∼N\left[0,1\right];X=\mathrm{XR}:\left[0∼1\right]$$
(2)$$X_{t}\left(\overset{n}{\underset{j=1}{\sum }}\;\frac{X_{ij}}{T\left(X_{tj}\right)}\right)t=1,2,3,\cdots ,$$

Among them, t = 1, 2, 3, …, *n*; t is the first-level variable, and *j* is the second-level variable.
(3)$$\begin{array}{cc} \;[X_{1}(\overset{5}{\underset{i=1}{\sum }}\;\frac{X_{1i}}{5})\;+ & X_{2}(\overset{3}{\underset{j=1}{\sum }}\;\frac{X_{2j}}{3})+X_{3}(\overset{4}{\underset{k=1}{\sum }}\;\frac{X_{3k}}{4})+X_{4}(\overset{4}{\underset{l=1}{\sum }}\;\frac{X_{4l}}{4})+X_{5}(\overset{3}{\underset{m=1}{\sum }}\;\frac{X_{5m}}{3})\\ & +X_{6}(\overset{4}{\underset{n=1}{\sum }}\;\frac{X_{6n}}{4})+X_{7}(\overset{5}{\underset{o=1}{\sum }}\;\frac{X_{7o}}{5})+X_{8}(\overset{6}{\underset{p=1}{\sum }}\;\frac{X_{8p}}{6})+X_{9}(\overset{4}{\underset{q=1}{\sum }}\;\frac{X_{9q}}{4})]\; \end{array}$$

The third step involves constructing the PMC surface. The PMC surface presents the evaluation results and advantages and disadvantages of policies in a three-dimensional image, allowing for an intuitive display of the results [41]. In this study, the evaluation samples selected are all publicly available policies, and *X*10 has no effect on the evaluation results. Therefore, *X*10 is removed to ensure the symmetry and balance of the matrix. The policy evaluation third-order matrix is constructed from *X*1–*X*9, and the surface plots of each policy are drawn using the calculation method shown in Equation (4).
(4)$$\mathrm{PMC}\mathrm{surface}=\left[\begin{array}{ccc} X_{1} & X_{2} & X_{3}\\ X_{4} & X_{5} & X_{6}\\ X_{7} & X_{8} & X_{9} \end{array}\right]$$

## 4. Results

### 4.1. Policy Bibliometrics

#### 4.1.1. Analysis of Word Frequency

In this study, the Rost CM6 software was used to perform word segmentation on 134 policy texts with a minimum length of two characters. Interference words were filtered out through manual judgment to remove irrelevant words such as adverbs and mood words. High-frequency words were extracted, and Table 5 shows the top 30 theme words ranked by frequency in the policy texts. Word Cloud was used to visualize the filtered high-frequency words into a word cloud, as shown in Figure 3.

The data shows that the three most frequently appearing words in the policy texts are *“*service*”* (8496 times), *“*medical*”* (7289 times), and *“*health*”* (5120 times), indicating that the main focus of China*’*s internet healthcare is on providing services and promoting health. The themes can be further categorized into four dimensions: entities, technology, management, and services.

Entities include the government, institutions, hospitals, and patients. One of the primary objectives of internet healthcare is to foster communication and collaboration between medical professionals and patients, thereby enhancing the quality and efficiency of healthcare services. The pivotal role played by these stakeholders is instrumental in achieving this objective. With the support of internet platforms, healthcare practitioners and patients can conveniently engage in activities such as medical consultations, appointment scheduling, and telemedicine, facilitating mutual interaction and collaboration between both parties.

Technology includes big data, internet, and artificial intelligence. One of the primary objectives of internet healthcare is to leverage technological advancements to improve the effectiveness of healthcare services and management. The continuous development and innovative applications of technology have facilitated the convenient and efficient acquisition, transmission, and processing of medical information. This enables healthcare professionals to access comprehensive medical knowledge and receive enhanced decision-making support. Simultaneously, patients benefit from more accessible healthcare services and gain access to sophisticated instruments for health management.

Management includes development, innovation, establishment, and improvement. One of the fundamental goals of internet healthcare is to establish a comprehensive management system that encompasses the development of policies and regulations, data security and privacy protection measures, as well as healthcare quality management protocols. Effective management plays a pivotal role in safeguarding the security and reliability of internet healthcare, while also enhancing the standardization and credibility of healthcare services. Moreover, it provides healthcare professionals and patients with a favorable healthcare experience, along with necessary safeguards.

Services include medical services, health, hygiene, diagnosis, and treatment. One of the primary goals of internet healthcare is to facilitate improved healthcare service experiences and enhanced convenience. Internet-based healthcare enables patients to access medical consultations, schedule appointments, and purchase medications at their convenience, regardless of time and location. Moreover, healthcare professionals can engage in activities such as remote diagnosis and tele-surgical guidance. These advancements significantly streamline medical communication and service provision between patients and healthcare providers, fostering greater accessibility and convenience in the healthcare delivery process.

In conclusion, the interconnected dimensions of stakeholders, technology, management, and services synergistically contribute to the attainment of the goals set forth in internet healthcare. These dimensions collectively propel the development and innovation of internet healthcare, facilitating the provision of convenient, efficient, and high-quality healthcare services. By strategically integrating and harmonizing these dimensions, the healthy advancement of internet healthcare can be fostered, resulting in heightened healthcare service quality and efficiency, as well as an improved healthcare experience and overall well-being for patients.

#### 4.1.2. Analysis of the Semantic Network

Furthermore, we conducted a co-occurrence analysis on the theme words to generate a co-occurrence matrix, revealing the degree of association between them, as shown in Table 6. A higher frequency of co-occurrence indicates a stronger correlation between the themes. We also utilized the net draw instrument to generate a semantic network graph, which provides a visual representation of the core content and connections of the policy text, as depicted in Figure 4.

From Figure 4, it can be seen that *“*Service*”* is at the core position and is a hot topic in Internet healthcare policies. It is closely related to key topics such as *“*Health*”*, *“*Development*”*, *“*Management*”*, *“*Strengthening*”*, and *“*Technology*”*. This indicates that the policy content mainly focuses on the service nature of internet healthcare services, emphasizes the innovation and application of medical technology, and strengthens and improves medical management. *“*Medical*”* is the foundation guarantee of internet healthcare policies, with related keywords such as *“*Institutions*”*, *“*Construction*”*, *“*Hospitals*”*, and *“*Guarantee*”*, reflecting that policy content revolves around the construction of medical institutions and medical guarantees. *“*Healthcare*”* is the direction emphasized in internet healthcare policies, with related words such as *“*Innovation*”*, *“*Improvement*”*, *“*Resources*”*, *“*Reform*”*, and *“*System*”*, reflecting that the policy*’*s main focus is on improving the healthcare system, innovation, and allocation of related resources. These keywords form a systematic focus of the policies, showing the content and structure of Internet healthcare policy texts as a whole.

### 4.2. Policy Instrument

Based on the classification of policy instruments mentioned above, specific policy items were taken as the unit of content analysis to code the text of internet healthcare policies, with the content analysis process carried out using the MAXQDA qualitative data analysis software. After the initial coding, multiple revisions were made to eliminate controversial coding terms. Eventually, 739 policy instrument coding units were obtained, and the classification results are presented in Table 7.
(1)Bias in policy supply and demand

Overall, China*’*s internet healthcare policies consider the use of supply-side, demand-side, and environmental policy instruments. However, there are significant differences in the extent to which these three policy instruments are utilized. Supply-side policy instruments are the most frequently used, with 362 measures accounting for 50.0% of the total and playing a leading role. Environmental policy instruments, with 304 measures accounting for 41.1%, are also commonly used. On the other hand, demand-side policy instruments are used less frequently, with only 73 measures, accounting for 9.9%, and they are relatively scattered. This overall reflects a *“*top-down*”* supply-side policy orientation dominated by the government, but the limited pull effect of demand-side policy instruments results in a significant deviation between supply and demand, which in turn affects the overall effectiveness of the development of internet healthcare.
(2)Uneven use of policy instruments

In practical application, there are differences in the use of the three types of policy instruments. Within the supply-side policy instruments, the government places a greater emphasis on information technology support (31.2%), public services (30.7%), and infrastructure construction (23.2%). This is because information technology is the fundamental support for the construction of the internet healthcare service system and plays an important role in improving the efficiency and quality of medical services. At the same time, the government also actively uses internet technology and platforms to provide medical and health services to the general public and vigorously strengthens the construction of infrastructure to ensure the interconnection of information and remove obstacles to the development of internet healthcare services. However, the application of talent cultivation (4.1%) and funding (1.4%) instruments is relatively low. Nevertheless, funding and talent are key factors in promoting the construction of the internet healthcare service system, and the absence of these two factors will affect the effectiveness of medical services, thereby restricting the development of Internet healthcare and health.

In terms of environmental policy instruments, the government has a greater focus on technical standards (46.4%) and legal regulation (32.9%). This reflects the fact that as a new concept, internet healthcare needs to be developed based on corresponding technical standards. Meanwhile, the government needs to adopt regulatory control to regulate and effectively supervise the behavior of various entities participating in internet healthcare services, ensuring the steady development of internet healthcare. The use of target planning instruments (11.8%) is the second most frequent, but the application of organizational coordination (5.6%) and publicity and promotion (3.3%) instruments is relatively low and neglected. To ensure the smooth promotion of internet healthcare, the government should strengthen the use of these two categories of demand policy instruments to create a favorable environment for the development of internet healthcare.

Among the demand-driven policy instruments, pilot standardization (42.5%) and medical insurance payment (30.1%) have received the most attention, indicating that the government attaches great importance to legislative pilot precedents and has gradually included internet healthcare services in the scope of medical insurance to drive the development of internet medicine. Public–private partnerships (23.3%) come next, but there is a significant gap in the use of international exchange instruments (4.1%). However, as internet healthcare has become an important global trend, international exchange and cooperation are crucial to promoting its development and progress. In the future, it is necessary to strengthen cooperation between countries and jointly promote the globalization of internet healthcare.
(3)Use of policy instruments in line with development history

In light of the stages of development in internet healthcare, as demonstrated in Table 8, there are notable distinctions in the utilization and allocation of policy instruments across various phases.

The usage of supply-side policy instruments is primarily concentrated during the initial construction phase, accounting for 67% of policy instrument utilization. This indicates that the government intends to increase resource inputs, such as manpower, materials, and finance, to promote the development of internet healthcare. Environmental policy instruments are mainly used during the standardization establishment phase and the explosive growth phase, with a utilization rate of 49% during both stages. The aim is to address legal and technical standard gaps, support internet healthcare in responding to development opportunities and risks, create a favorable policy environment, and indirectly promote its development. The utilization of standard and regulatory instruments increases for each phase, promoting the standardization and rationalization of internet healthcare services. The utilization of demand-driven policy instruments has a small variation in percentage across the three stages, accounting for 10%, 12%, and 8% during the initial construction phase, standardization establishment phase, and explosive growth phase, respectively. The utilization of medical insurance payment instruments increases during each phase, promoting the sustainable development of internet healthcare services. It is evident that the government has made a rational selection of policy instruments to address the development requirements and policy demands of different phases when formulating internet healthcare policies.

### 4.3. PMC Index Evaluation

#### 4.3.1. Selection of the Sample

The PMC Index Model is designed for analyzing and studying specialized policies, while policies related to internet healthcare are mostly comprehensive, with provisions dispersed across comprehensive documents such as *“*Internet Plus*”*, deepening medical and health system reform, and the 13th and 14th Five-Year Plans, among others. This study further screened the collected policy documents and selected 16 specialized representative policies as evaluation samples to ensure the reasonableness and comprehensiveness of the policy sample, as shown in Appendix B.

#### 4.3.2. PMC Index Calculation and Surface Construction

According to the previously established steps, the PMC index scores for each policy were calculated, and the scores were sorted and classified into levels, as shown in Table 9. Surface plots were generated through three-dimensional visualization to demonstrate the evaluation scores and strengths and weaknesses of policies. The presence of surface undulations and color variations indicates differences in policy scores. Raised areas correspond to higher policy evaluation scores, while recessed areas correspond to lower policy evaluation scores. This study presents the PMC surfaces of the policy with the highest PMC index score (P4) and the lowest score (P2), as shown in Figure 5.

#### 4.3.3. Analysis of Evaluation Results


(1)Longitudinal evaluation


According to the PMC index results in Table 9, it can be seen that all policies have a membership level of either excellent or qualified. The mean PMC index of the 16 policies is 7.31. The order of policy scores from high to low is P4 > P1 > P8 > P16 > P3 > P5 > P6 > P14 > P7 > P11 > P15 > P13 > P12 > P10 > P9 > P2.

Grade A excellent policies are policies with a score between 7 to 8.99 points and a policy grade of excellent. They include P1, P3, P4, P5, P6, P7, P8, P11, P14, and P16. For instance, P4 has a PMC index of 8.75; this policy is an opinion on the development of health and medical big data application issued by the State Council, ranking first. The PMC surface plot depicted in Figure 5a showcases a predominance of convex surfaces, indicating that P4 scored higher across a range of indicators. In terms of specific scores, 8 out of 10 primary variables received the highest score, indicating that the policy*’*s overall design is scientifically reasonable and its content covers all indicators in the evaluation model. To further enhance this policy, optimization of the policy level (*X*4) increased cross-departmental collaboration and supervision and ensured the policy*’*s effectiveness and implementation could be considered to promote the development of medical big data.

Grade B qualified policies are those with a score ranging from 5 to 6.99 points and a policy grade of qualified. They include P2, P9, P10, P12, P13, and P15. For example, P2 has a PMC index of 5.65. This policy is a set of opinions issued by the National Health and Family Planning Commission regarding the promotion of telemedicine services in healthcare institutions, ranking sixteenth. The PMC surface plot depicted in Figure 5b shows fewer convex points on the surface plot, and most of the surfaces are concave, indicating that P2 received lower scores across various indicators, reflecting its relative deficiencies compared to other policies. In terms of specific scores, 9 out of 10 primary variables require further improvement, with the policy objectives (*X*5) score significantly lower than the average. Therefore, the scope of policy recipients needs to be expanded further. Furthermore, the scores for policy instruments (*X*2), policy goals (*X*6), policy functions (*X*7), incentives and constraints (*X*8), and policy evaluation (*X*9) are also low. This indicates that the policy*’*s mode of action is relatively single, and its coverage is relatively narrow. It is recommended to implement a multi-stage and well-planned approach to address these concerns. A more scientific and effective improvement path can be pursued by considering the difference between variable scores and the mean value. The suggested path for improvement is as follows: *X*5→*X*9→*X*8→*X*7→*X*2→*X*6→*X*4→*X*1→*X*3.
(2)Cross-sectional evaluation

This paper generated a radar chart based on the scores of their primary indicators to facilitate the comparison and evaluation of the selected 16 policies, as illustrated in Figure 6.

The results indicate that overall, the policies performed well, but there are still some shortcomings in individual indicators. Looking at the average scores of each indicator, *X*1, *X*2, *X*5, *X*6, and *X*10 have relatively high average scores, while *X*3 and *X*4 have the lowest average scores. Specifically, the average score for policy nature (*X*1) is 0.84, indicating that most policies focused on description, suggestion, guidance, and supervision but had less involvement from a predictive perspective. The average score for policy instruments (*X*2) is 0.84, with supply-side and environmental policy instruments being widely used in most policies but demand-side policy instruments being less frequently used. The average score for policy effectiveness (*X*3) is 0.28, indicating that most policies had only short-term effects, suggesting that the development target deadline for internet healthcare policies is relatively narrow. The average score for policy level (*X*4) is 0.3, indicating that most policies are issued by single-type entities, and cooperation between different levels of departments needs to be strengthened. The average score for policy objects (*X*5) is 0.87, with all policies involving the participation of medical institutions and government agencies, but some policies lack cooperation with third-party institutions. The average score for policy goals (*X*6) is 0.86, with all policies promoting health and implementing corresponding measures to optimize services, but some policies lack measures for standard setting and resource balance. The average score for policy functions (*X*7) is 0.8, with all policies including functions for service optimization, supervision and constraints, and normative guidance, but some policies lack functions for innovation-driven and industry support. The average score for incentive constraints (*X*8) is 0.71, with areas for improvement in talent development and funding, especially in the lack of specific measures for funding. The average score for policy evaluation (*X*9) is 0.81, with all policies based on sufficient evidence and scientific programs but with relatively thin descriptions of objectives and content, needing to strengthen this aspect of expression. The average score for policy openness (*X*10) is 1, indicating that all policies were open policies.

## 5. Conclusions

This study has established a framework of “policy keywords—policy instruments—policy evaluation” to analyze and evaluate the internet healthcare policies in China from a multidimensional perspective, as shown in Table 10. The research findings demonstrate that the content of internet healthcare policies primarily revolves around the aspects of user interaction, service quality, technological innovation, and regulatory standards, aiming to promote the development of internet healthcare and enhance its efficiency and quality. However, the usage of policy instruments in internet healthcare policies generally aligns with the overall development trajectory of internet healthcare in China but also exhibits an imbalanced pattern and lacks complementarity. Among the policy instruments used, there is a greater emphasis on supply-side and environmental instruments, while the attention given to demand-side instruments is insufficient, thus slowing down the progress of “Internet healthcare.” Additionally, there are significant gaps in the supply of instruments related to funding investment and talent development, deficiencies in environmental aspects such as promotion and organizational coordination, and a lack of international exchange and cooperation on the demand side. The analysis using the PMC Index Model indicates an average PMC index of 7.14, reflecting a generally favorable situation of internet healthcare policies. However, there exist variations in the effectiveness of different policies, and some policy provisions require improvement in terms of efficacy, objectives, and functions, highlighting the need for strengthening and enhancing the policies.

Furthermore, it is recognized that internet healthcare policies worldwide encounter common challenges, such as imbalanced use of policy instruments, weak complementarity among policies, and inadequate policy design due to rapid technological advancements. These challenges impede the effective regulation of online healthcare services.

Based on the aforementioned analysis, the following policy recommendations are proposed:(1)Achieving a balanced utilization of policy instruments is crucial. Particularly, it is essential to enhance the combination and application of demand-side policy instruments, as they directly stimulate policy actors and yield significant effects [42]. Considering the constraints posed by limited public fiscal revenue in China, policies should be employed as the primary means to propel the development of internet healthcare. This can be achieved by attracting increased participation from various social forces in areas and segments aligned with intelligent healthcare and online medical services, thereby alleviating the policy’s financial, technological, and human resource burdens [43]. Furthermore, policy formulation should emphasize the importance of knowledge exchange and cooperation among different nations in the realm of internet healthcare, facilitating resource integration and sharing to collectively foster the advancement of internet healthcare endeavors;(2)Strengthen talent cultivation and fiscal investment. In the context of future policy formulation, it is crucial to reinforce the mechanisms for talent cultivation and professional development in the realm of internet healthcare. Particular emphasis should be placed on nurturing highly skilled individuals capable of effectively integrating knowledge in internet technologies and medical domains. Furthermore, attention ought to be directed toward augmenting governmental fiscal investment in the establishment of internet healthcare service systems. Encouragement and support should be extended to healthcare institutions at various levels, empowering them to leverage pertinent information technologies, notably the internet, to deliver healthcare services, thus fostering enhancements in service quality and capacity. The government is urged to intensify its efforts in amplifying the outreach and promotion of policies to ensure the smooth progress of internet healthcare [44]. This necessitates the broad utilization of diverse communication channels to enhance the dissemination and interpretation of internet healthcare policies. Additionally, a coordinated approach involving pertinent departments must be fostered, engendering collaborative mechanisms that engender an enabling environment for the development of internet healthcare;(3)Undertake an integrated and systemic evaluation of internet healthcare policies. In the realm of policy formulation, it is imperative to establish a robust mechanism for conducting comprehensive and systemic evaluations of internet healthcare policies. Due emphasis should be placed on developing an evaluative framework that encompasses the holistic nature of policies moving forward, ensuring a thorough assessment of their efficacy, objectives, and functionalities. This approach safeguards the integrity, scientific rigor, and feasibility of the policies. At present, as policies in the domain of internet healthcare undergo refinement, it is crucial to accord significant importance to long-term planning, laying a solid foundation for sustainable growth. Furthermore, policy improvement necessitates expanded coverage that enhances the functionality of the policies. Within the ambit of long-term planning, it becomes indispensable to align strategic objectives with attainable medium- and short-term targets, supplemented by periodic evaluations of policy outcomes at different stages.

Lastly, due to the predominantly macroscopic perspective adopted during policy formulation, certain provisions within the policies may possess a high degree of generality, lacking specific guiding principles. As a result, the practical effectiveness of policy implementation may be suboptimal. Future policy implementation endeavors should entail further elaboration of execution plans by policymakers to address this. This refinement process should prioritize ensuring the policy’s comprehensibility and operational feasibility. Concurrently, policy executors should diligently formulate corresponding implementation schemes and strategies that align with specific requirements and guidelines. This approach serves to facilitate the seamless execution and effective implementation of policies. Furthermore, augmenting oversight and evaluation mechanisms throughout the policy execution process is imperative [45]. This proactive approach allows for the timely identification of challenges and areas requiring improvement, thereby augmenting the efficacy of policy implementation.

## Figures and Tables

**Figure 1 healthcare-11-01905-f001:**
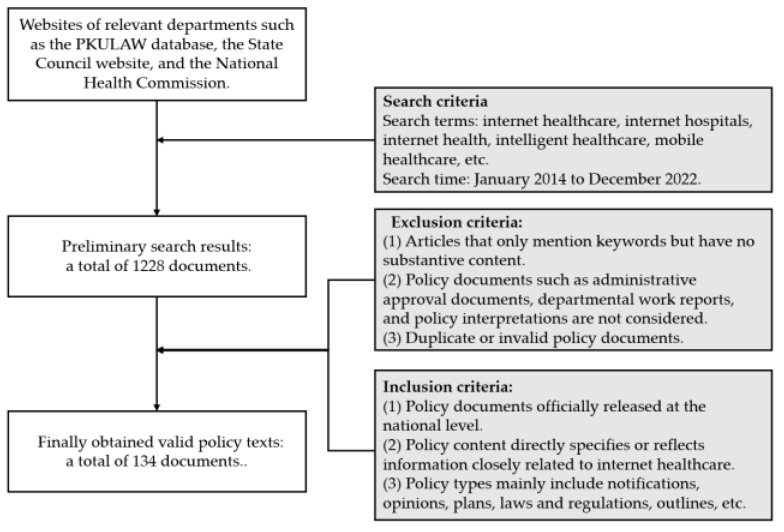
Policy text data collection process.

**Figure 2 healthcare-11-01905-f002:**
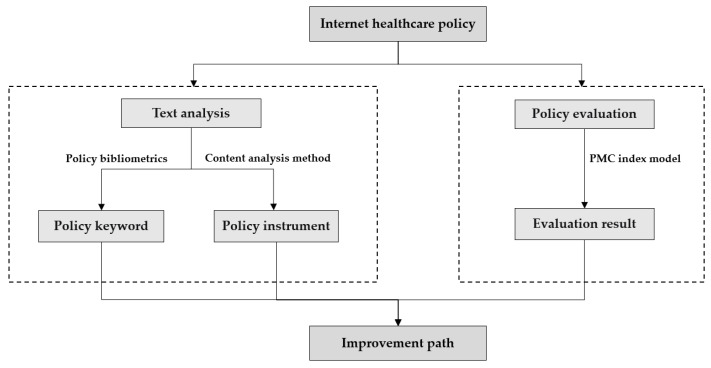
Research framework.

**Figure 3 healthcare-11-01905-f003:**
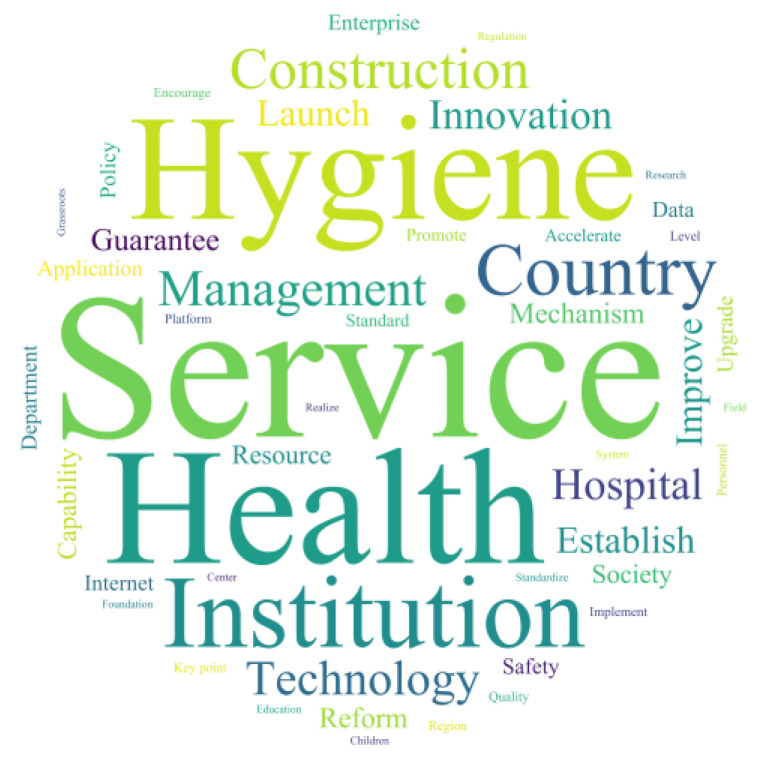
Policy text word cloud.

**Figure 4 healthcare-11-01905-f004:**
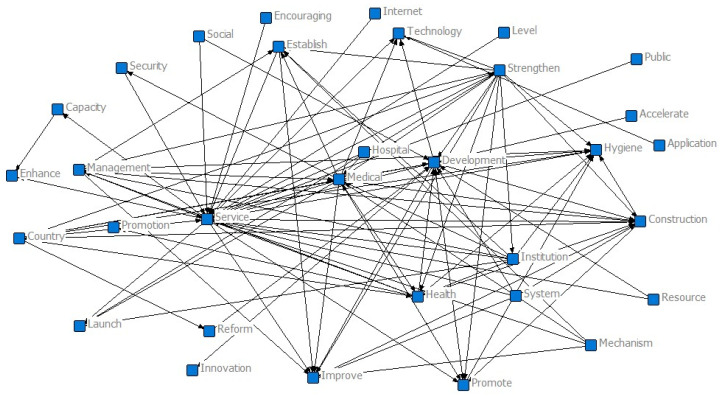
Semantic network of the internet healthcare policy.

**Figure 5 healthcare-11-01905-f005:**
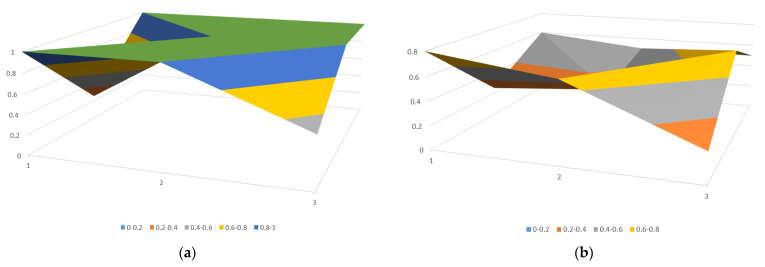
(**a**) PMC surface of P4, and (**b**) PMC surface of P2.

**Figure 6 healthcare-11-01905-f006:**
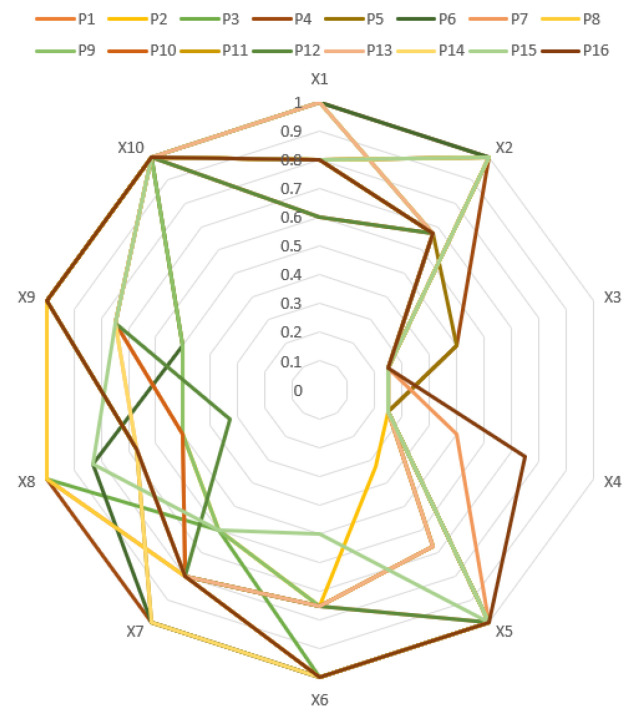
Radar chart of first-level indicator scores.

**Table 1 healthcare-11-01905-t001:** Summary of the literature on internet healthcare policies research.

Research Topic	Related Work	Pros	Cons
Study of the policy development system	Reviewing the current status and evolution of internet healthcare policies	Uncover the background, environment, and significance inherent in policy changes and explore the development and changes in internet healthcare	Limited research perspective, relying on superficial qualitative descriptive analysis
Evaluation of the implementation effectiveness of internet healthcare policies	Studying internet hospitals and evaluating the implementation effectiveness of internet healthcare policies	Examine the scientific and feasibility of internet healthcare policies in light of the actual development of internet healthcare	Inadequate comprehensiveness of research content and excessive subjectivity in evaluation
Analysis of factors influencing the implementation of internet healthcare policies	Using empirical analysis to explore the factors influencing the implementation of internet healthcare policies	Enhanced understanding and better response to various factors involved in policy formulation and implementation	More focus on the main subjects of internet healthcare policies and some external environmental factors, insufficient emphasis on policy content

**Table 2 healthcare-11-01905-t002:** Classification and meaning of internet healthcare policy instrument.

Instrument Type	Specific Name	Meaning
Supply	Infrastructure construction	Regulate the relevant infrastructure construction for the development and application of internet healthcare.
	Talent cultivation	Various education and training activities for practitioners of internet healthcare strengthen the training of relevant talents.
	Fund investment	Provide financial and technological support, including setting up special funds and subsidies, etc.
	Public services	Promote the integration of the internet and medical healthcare services, improving the internet healthcare service system.
	Information technology support	Build relevant databases and knowledge bases, utilizing modern information technology to provide information exchange and medical services.
	Resource allocation	Government macro-control of resources, coordinating resource allocation, and promoting equalization of basic public health.
Environment	Target planning	Establish macro targets and overall planning, determining the development direction of internet healthcare.
	Legal regulation	Normative and regulatory measures on the behavior of medical service providers through laws and systems ensure orderly development.
	Technical standards	Construct the relevant technical standards and evaluation system of internet healthcare services.
	Promotion	Active promotion of the policy measures and progress of internet healthcare, generating positive public opinion.
	Organizational coordination	The government establishes a special leadership organization to coordinate the management of internet healthcare services and promote overall planning.
Demand	Pilot standardization	Conduct pilot projects and standardization of internet healthcare service projects.
	Medical insurance payment	Promote the reform of internet diagnosis, treatment, and payment models, expanding the medical insurance coverage of internet healthcare.
	International exchanges	Exchange knowledge and services of internet healthcare with other countries to learn from advanced foreign experiences.
	Public–private partnerships	The government encourages and guides social capital to participate in cooperation.

**Table 3 healthcare-11-01905-t003:** Evaluation index and standard setting for internet healthcare policies.

First-Level Variables	Second-Level Variables	Evaluation Standards
X1 Policy nature	X1:1 Predictive X1:2 Advisory X1:3 Descriptive X1:4 Guiding X1:5 Regulatory	Judge whether the policy content has predictive, advisory, descriptive, guiding, and regulatory characteristics. If yes, the score is 1; otherwise, the score is 0.
X2 Policy instruments	X2:1 Supply-side X2:2 Demand-side X2:3 Environment-side	Judge whether the policy instruments are supply-side, demand-side, or environment-side. If yes, the score is 1; otherwise, the score is 0.
X3 Policy effectiveness [36]	X3:1 Temporary X3:2 Short-term X3:3 Medium-term X3:4 Long-term	Judge whether the policy impact is within one year, 1–5 years, 6–10 years, or more than 10 years. If yes, the score is 1; otherwise, the score is 0.
X4 Policy level [37]	X4:1 State Council X4:2 State Council departments X4:3 State Council-affiliated institutions X4:4 National bureaus managed by departments	Judge whether the policy issuer is the State Council, State Council departments, State Council-affiliated institutions, or national bureaus managed by departments. If yes, the score is 1; otherwise, the score is 0.
X5 Policy objectives	X5:1 Government institutions X5:2 Medical institutions X5:3 Third-party institutions	Judge whether the policy objectives include government institutions, medical institutions, or third-party institutions. If yes, the score is 1; otherwise, the score is 0.
X6 Policy goals	X6:1 Health X6:2 Services X6:3 Resources X6:4 Standards	Judge whether the policy goals are to promote health, optimize services, balance resources, or establish standards. If yes, the score is 1; otherwise, the score is 0.
X7 Policy functions [38]	X7:1 Regulatory constraints X7:2 Normative guidance X7:3 Service optimization X7:4 Innovation-driven X7:5 Industrial support	Judge whether the policy has functions of regulatory constraints, normative guidance, service optimization, innovation-driven, or industrial support. If yes, the score is 1; otherwise, the score is 0.
X8 Incentives and constraints [39]	X8:1 Legal protection X8:2 Supervision and assessment X8:3 Skill training X8:4 Talent introduction X8:5 Capital investment X8:6 Organizational implementation	Judge whether incentives and constraints involve legal protection, supervision, assessment, skill training, talent introduction, capital investment, or organizational implementation. If yes, the score is 1; otherwise, the score is 0.
X9 Policy evaluation [40]	X9:1 Sufficient evidence X9:2 Clear objectives X9:3 Scientific plan X9:4 Detailed content	Judge whether the policy evaluation is based on sufficient evidence, and has clear objectives, a scientific plan, and detailed content. If yes, the score is 1; otherwise, the score is 0.
X10 Policy transparency	—	Judge whether the policy is transparent. If yes, the score is 1; otherwise, the score is 0.

**Table 4 healthcare-11-01905-t004:** Policy PMC index evaluation level.

PMC Score	9–10	7–8.99	5–6.99	0–4.99
Evaluation Level	Optimal	Excellent	Qualified	Not Up to Standard

**Table 5 healthcare-11-01905-t005:** High-frequency policy themes (excerpt).

Keyword	Frequency	Keyword	Frequency
Service	8496	Launch	2112
Medical	7289	Improve	2069
Health	5120	Guarantee	2010
Development	4687	Mechanism	1864
Hygiene	4603	Reform	1838
Institution	3931	Resource	1816
Country	3383	Society	1783
Construction	3312	Promote	1750
Strengthen	3301	Capability	1707
Management	3270	System	1706
Technology	2873	Application	1691
Promote	2351	Department	1601
Hospital	2336	Safety	1585
Innovation	2316	Internet	1562
Establish	2179	Enterprise	1561

**Table 6 healthcare-11-01905-t006:** Co-occurrence matrix (excerpt).

Keyword	Service	Medical	Health	Development	Hygiene	Institution	Country	Construction	Strengthen	Management
Service	8496	1878	1188	1111	1289	1273	681	990	1027	1078
Medical	1878	7289	1169	756	1491	1610	590	704	882	947
Health	1188	1169	5120	833	1448	690	805	666	724	712
Development	1111	756	833	4687	691	0	898	811	722	0
Healthcare	1289	1491	1448	691	4603	1068	920	648	743	703
Institution	1273	1610	690	0	1068	3931	0	0	676	704
Country	681	590	805	898	920	0	3383	587	558	0
Construction	990	704	666	811	648	0	587	3312	973	632
Strengthen	1027	882	724	722	743	676	558	973	3301	800
Management	1078	947	712	0	703	704	0	632	800	3270

**Table 7 healthcare-11-01905-t007:** Distribution of various policy instruments.

Instrument Type	Proportion	Specific Instrument	Frequency	Proportion
Supply (326)	50%	Information technology support	113	31.2%
		Public services	111	30.7%
		Infrastructure construction	84	23.2%
		Resource allocation	34	9.4%
		Talent cultivation	15	4.1%
		Fund investment	5	1.4%
Environment (304)	41.1%	Technical standards	141	46.4%
		Legal regulation	100	32.9%
		Target planning	36	11.8%
		Organizational coordination	17	5.6%
		Promotion	10	3.3%
Demand (73)	9.9%	Pilot standardization	31	42.5%
		Medical insurance payment	22	30.1%
		Public-private partnerships	17	23.3%
		International exchanges	3	4.1%

**Table 8 healthcare-11-01905-t008:** Distribution of policy instruments in different development stages.

Instrument Type	Specific Instrument	2014–2017	2018–2019	2020–2022
Supply	Infrastructure construction	29	26	27
	Resource allocation	18	8	8
	Public services	39	35	37
	Information technology support	58	15	42
	Fund investment	2	2	1
	Talent cultivation	4	7	4
Environment	Technical standards	22	39	80
	Legal regulation	10	56	34
	Target planning	14	16	6
	Organizational coordination	5	4	8
	Promotion	1	3	6
Demand	Medical insurance payment	2	9	11
	Pilot standardization	9	13	9
	International exchanges	2	0	1
	Public–private partnerships	9	6	2

**Table 9 healthcare-11-01905-t009:** PMC index of 16 internet healthcare policies.

	*X*1	*X*2	*X*3	*X*4	*X*5	*X*6	*X*7	*X*8	*X*9	*X*10	PMC Index	Rating	Ranking
P1	0.8	1	0.25	0.25	1	1	0.8	1	1	1	8.1	Excellent	2
P2	0.8	0.67	0.25	0.25	0.33	0.75	0.6	0.5	0.5	1	5.65	Qualified	16
P3	0.8	1	0.25	0.25	1	1	0.6	1	1	1	7.9	Excellent	5
P4	1	1	0.5	0.25	1	1	1	1	1	1	8.75	Excellent	1
P5	0.8	0.67	0.5	0.25	1	1	1	0.67	1	1	7.89	Excellent	6
P6	1	1	0.25	0.25	1	1	1	0.83	0.5	1	7.83	Excellent	7
P7	0.8	1	0.25	0.5	1	0.75	0.8	0.5	0.75	1	7.35	Excellent	9
P8	1	0.67	0.25	0.25	1	1	0.8	1	1	1	7.97	Excellent	3
P9	1	0.67	0.25	0.25	0.67	0.75	0.6	0.5	0.5	1	6.19	Qualified	15
P10	0.6	0.67	0.25	0.25	0.67	0.75	0.8	0.5	0.75	1	6.24	Qualified	14
P11	0.8	1	0.25	0.25	0.67	0.75	0.8	0.67	1	1	7.19	Excellent	10
P12	0.6	0.67	0.25	0.25	1	0.75	0.8	0.33	0.75	1	6.4	Qualified	13
P13	1	0.67	0.25	0.25	0.67	0.75	0.8	0.67	0.75	1	6.81	Qualified	12
P14	0.8	1	0.25	0.25	1	1	1	0.67	0.75	1	7.72	Excellent	8
P15	0.8	1	0.25	0.25	1	0.5	0.6	0.83	0.75	1	6.98	Qualified	11
P16	0.8	0.67	0.25	0.75	1	1	0.8	0.67	1	1	7.94	Excellent	4
Average	0.84	0.84	0.28	0.30	0.88	0.86	0.8	0.71	0.81	1	7.30		

**Table 10 healthcare-11-01905-t010:** Summary of research findings.

Research Perspective	Research Work	Finding
Policy keywords	This study utilized the policy literature’s bibliometrics to conduct word frequency statistics and semantic network analysis on the topic keywords.	The content of internet healthcare policies primarily revolves around the aspects of user interaction, service quality, technological innovation, and regulatory standards.
Policy instruments	This study employed content analysis to analyze the supply, demand, and environmental characteristics of policies.	The usage of policy instruments generally aligns with the overall development trajectory of Internet healthcare but also exhibits an imbalanced pattern and lacks complementarity.
Policy evaluation	This study constructed a PMC index evaluation model to identify deficiencies in policy design across various dimensions	The policies are generally well-formulated, but there are still some shortcomings in individual indicators.

## Data Availability

The data used to support the findings of this study are available from the corresponding author upon request.

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
