# Peer review of "Internet Healthcare Policy Analysis, Evaluation, and Improvement Path: Multidimensional Perspectives"

_healthcare, 2023, doi:10.3390/healthcare11131905_

Round 1
Reviewer 1 Report
Authors present with clarity the method they used and the associated analysis of the results.
Some comments below:
1. Increase the resolutions of the figures
2. Define accurately the objectives of this study in Introduction
3. Shorten the conclusion and put the findings in a different section above the conclusions.
4 In this section, create a table where all these findings are depicted. How do these findings are compared with similar findings in other countries?
5. In literature review, enter a table where you summarize the related works and show the pros and cons or the findings to illustrate the comparison with you work.
Authors should check the document for English and typos.
Author Response
Thank you for providing us with the opportunity to revise and resubmit our manuscript. We have made substantial revisions and every attempt to incorporate the your suggestions.
Please see the attachment.

Reviewer 2 Report
Contribution/Summary: This research study analyzes and evaluates Chinese internet healthcare policy materials to uncover concerns and deficiencies. Policy bibliometrics, content analysis, and the PMC index model are used to analyze 134 national policy documents. The study found that Chinese internet healthcare policies prioritize service quality, technological innovation, and management uniformity.
Comments/Suggestions: 1. The study article's core contribution and key findings are succinctly summarized in the abstract, which is generally well-written. Nevertheless, one enhancement recommendation would be to give more particular information regarding the study's methodology, as doing so would aid readers in understanding the research techniques used. For instance, the introduction may offer more details regarding the PMC index model and briefly clarify the specific policy bibliometrics and content analysis techniques utilized to examine the relevant policy papers. 2. The introduction may give more background information on the significance of governmental guidelines for the growth of Internet healthcare, particularly in light of China's healthcare industry. 3. In the introduction, it would be useful if you could include a list of important points that summarize your findings to help readers better comprehend the primary contributions of your work. 4. In addition, it would be useful to add a short paragraph at the end of the introduction that describes the structure of the rest of the paper. 5. Section 3.2 might better describe the rationale for combining the three viewpoints of topic keywords, policy instruments, and policy assessment, as well as how each contributes to the overall analysis of internet healthcare policies in China. 6. It would be beneficial if the authors could include a paragraph discussing the usage of formal methods in the healthcare industry, since this might provide additional insights into the possibilities for employing rigorous and systematic approaches to analyze and optimize healthcare policies.7. For this purpose the authors may consider including the following references (And others): a. https://link.springer.com/chapter/10.1007/978-3-030-51517-1_33 b. https://link.springer.com/chapter/10.1007/978-3-319-12214-4_15 8. The advantages of using policy bibliometrics as a research methodology may be explained in Section 3.3.1 in a more clear and concise manner, as could how it relates to existing policy research paradigms.
9. It would be helpful if Section 3.3.3 provided additional detailed details concerning the process of selecting variables and parameters, as well as the selection procedure for the 10 first-level variables and the 38 second-level variables. 10. The four aspects of entities, technology, management, and services, as well as how they connect to the fundamental objective of China's internet healthcare policy, might be explained in greater detail in Section 4.1.1.
11. It would be helpful if Section 4.3 provided a clearer explanation of how the generated PMC surfaces help highlight the strengths and weaknesses of policies and how this knowledge may be used to improve the effectiveness of policies.
13. In addition, the conclusion should do a better job of properly summarizing the most important recommendations for improving China's internet healthcare policies and how those recommendations relate to the issues that have been discovered.
14. Additionally, the conclusion might provide a more explicit explanation of how the proposals match with China's national strategic goals for growing internet healthcare, as well as how they could inform the creation of future policies.
15. Lastly, the conclusion might give more particular details about the potential opportunities and challenges connected with adopting the recommended policy changes, as well as how such obstacles and opportunities could be managed.
Can be improved.
Author Response

(The authors gave the same response as above.)

Reviewer 3 Report
The authors touched upon a very timely and interesting topic. The field of study is new, the method is solid, and the implications are profound. Overall I think this paper has significance for the current knowledge, and I have some monir recommendations for the authors to consider so that the significance of this piece is more straightforward.
1. The introduction could be revised so the research question and research objectives are stated earlier. With that said, I am also a bit lost about the focus of this paper - what aspects of internet healthcare are you analyzing and assessing? Are you just giving a review of the policy landscape, or are you making recommendations on certain policy deficiencies that you plan to identify from the analyses?
2. In 3.1, why the policy retrieval period was limited to 2014-2022? Although the initial construction phase is 2014-2017, China had a round of healthcare reform in 2013 which encourages the private sector’s participation to reform the stymied public hospital system, and another health reform in 2015 that relinquishes the kickbacks from selling drugs in hospitals. How do you plan to consider these healthcare reform policies in your thinking? You need to articulate these in the manuscript.
3. Why do the authors only consider national policy? I understand this as central policies are implemented by local governments. There could be central-local conflicts in policy implementation. If the authors have reasons for only using national policies, they should be stated in the paper.
4. I am looking for more Insights from the discussion and conclusion. The policy instrument analysis in 5.1 is plain. It is easy to say that all the supply-, demand-side, and environmental policy instruments should be improved, but the hard fact is that “you cannot have it all”. In your opinion, which one should be prioritized? Also, do the authors perceive any conflicts or contradictions? In addition, in conclusion, you can also add one paragraph to summarize your findings from the results. The current format of going straight to policy recommendations is misleading.
Author Response
Thank you for providing us with the opportunity to revise and resubmit our manuscript. We have made substantial revisions and every attempt to incorporate your suggestions.
Please see the attachment.

Round 2
Reviewer 2 Report
The authors considered my comments and suggestions. Good luck.
May be improved
Reviewer 3 Report
The authors have addressed my comments and concerns.